# Development and Field Validation of Lidocaine-Loaded Castration Bands for Bovine Pain Mitigation

**DOI:** 10.3390/ani10122363

**Published:** 2020-12-10

**Authors:** James W. Saville, Joseph A. Ross, Tyler Trefz, Crystal Schatz, Heather Matheson-Bird, Brenda Ralston, Ori Granot, Karin Schmid, Richard Terry, Nicholas D. Allan, Jeremy E. Wulff, Merle Olson

**Affiliations:** 1Department of Chemistry, University of Victoria, Victoria, BC V8W 3V6, Canada; jameswsaville@gmail.com (J.W.S.); ttrefz@uvic.ca (T.T.); origuvic@gmail.com (O.G.); 2Chinook Contract Research Inc., Airdrie, AB T4A 0C3, Canada; joe.ross@ccr01.com (J.A.R.); crystal.schatz@ccr01.com (C.S.); heather@ccr01.com (H.M.-B.); 3Alberta Agriculture and Forestry, Airdrie, AB T4A 0C3, Canada; Brenda.Ralston@gov.ab.ca; 4Alberta Beef Producers, Calgary, AB T2E 7H7, Canada; KarinS@albertabeef.org; 5Richard Terry Innovations, LLC, Conyers, GA 30094, USA; rt33@bellsouth.net; 6Alberta Veterinary Laboratories, Calgary, AB T2C 5N6, Canada; merle.olson@avetlabs.com

**Keywords:** animal welfare, bovine, castration, elastration, lidocaine, pain mitigation

## Abstract

**Simple Summary:**

Castration is among the most common management procedures performed in the dairy and beef cattle industries. Despite the widely accepted benefits of castration, all castration methods produce pain and distress. While slower than other approaches, castration by banding is simple, inexpensive, and produces fewer complications. We have therefore focused the present study on developing herd-level pain mitigation during banded castration. Because lidocaine is effective at reducing pain and can be delivered topically, we have formulated lidocaine-loaded castration bands (LLBs) to deliver local pain relief to calves and bulls during banded castration. Laboratory results indicated a rapid release of lidocaine for the first 30 min, followed by a slow release for at least 48 h. Field studies indicated that, for both lidocaine delivery into tissues and pain mitigation, LLBs performed at least as well as standard lidocaine injections in the short term and outperformed lidocaine injections in the long term. LLBs delivered therapeutic quantities of lidocaine into scrotal tissues over a period of at least seven days in cattle. This approach would provide long-term pain mitigation to the animals and, by avoiding surgery or the administration of injections, would also decrease the time and handling costs for the producer.

**Abstract:**

Castration is among the most common management procedures performed in the dairy and beef cattle industries and is mainly performed by surgery or elastic banding. Despite the various benefits of castration, all methods produce pain and distress. Castration by banding is simple, inexpensive, produces fewer complications, and can be performed in a high-throughput manner. Because lidocaine, a local anesthetic, can be delivered to trauma sites topically, we have formulated lidocaine-loaded castration bands (LLBs) to deliver local pain relief to calves during banded castration. The initial lidocaine content of three band types developed was between 80 and 200 mg per band. The transfer kinetics of lidocaine into tissue was determined in vitro, indicating a rapid release for the first 30 min, followed by a slow release lasting at least 48 h. Furthermore, the lidocaine delivery and pain mitigation effects of these LLBs were compared to standard lidocaine injections in vivo. Field studies indicated that LLBs performed at least as well as lidocaine injections for short-term lidocaine delivery into tissues and pain mitigation. Moreover, LLBs significantly outperformed lidocaine injections for long-term delivery and pain mitigation. The concentrations of lidocaine in the LLB-treated tissue samples were generally in the range of 0.5–3.5 mg of lidocaine per gram of tissue and were overall highest after 6 h. Lidocaine-loaded elastration bands deliver therapeutic quantities of lidocaine into scrotal tissues over a period of at least seven days in cattle. This approach would provide long-term pain mitigation to the animals and, by avoiding surgery or the administration of injections, would also decrease the time and handling costs for the producer.

## 1. Introduction

Castration is one of the most common management procedures performed in the dairy and beef cattle industries. In the United States alone, an estimated 15 million bovine castration procedures are performed each year [1]. In modern agriculture, castration is undertaken to: (i) decrease aggression, enhancing safety for handlers and animals; (ii) selectively control livestock populations; and (iii) produce higher-quality meats that command higher market prices [1,2,3]. Despite the widely accepted benefits of castration, all methods of castration result in physiological, neuroendocrine, and behavioural changes indicative of distress and pain [1,2,3,4,5,6]. The Canadian Code of Practice for the Care and Handling of Beef Cattle has recently been updated, advising that pain control be used for castration of bulls older than 6 months, and it is expected that pain management will be advised for all castrations in the near future [7]. Pain management is important for the well-being of the animal and castration recovery time [3,4]. Existing techniques (i.e., ring blocking the scrotum or spermatic cord injection) to deliver anesthesia during castration are not practical at the herd level [7].

Currently, only one pain control product (Meloxicam) is labeled for alleviating the pain associated with castration. This product’s label claim is for up to 56 h of pain relief, requiring approximately 2.5 h to reach maximal plasma concentration [5]; however, given the practicalities of cattle handling, and the desire of producers to mitigate the stress of handling, it is often administered at the time of castration. This means that pain control does not begin until sometime after the initial castration procedure and wears off well before the wounds have healed. Accordingly, producers, scientists, and agriculture ministries are seeking practical solutions to the challenge of herd-level pain mitigation for castration.

Many methods of castration are employed by the cattle industry, each with inherent advantages, disadvantages, and unique considerations. Surgical testicle removal is a common means of castration but is losing popularity [1]. Surgical castration is rapid and effective; however, in certain jurisdictions, a licensed veterinarian may be required on site to perform this procedure [8]. Complications are not uncommon and include hemorrhage, infection, and myiasis. Castration by banding (Figure 1) is an alternative technique and the second-most common method applied [1]. Banded castration takes 3–6 weeks before the testes are sloughed off, making it the slowest means of castration [1,9]. Although there are fewer complications with banded castration, it produces chronic pain that can last at least 42 days [10]. Additionally, castration band application is simple and may be performed in a high-throughput manner [1,9]. We have therefore focused this study on developing herd-level pain mitigation for banded bovine castrations of different ages and body weights.

Local anesthetics are used to numb acute, chronic, and surgical pain across all species. Anesthetics decrease pain sensation by preventing the transmission of nerve impulses from the site of trauma to the brain. Lidocaine is an archetypal local anesthetic, characterized by its rapid onset and prolonged duration of effect [11]. Meléndez et al. (2018) found that injected lidocaine was effective at reducing physiological and behavioural indicators of pain (measured by salivary cortisol, white blood cell counts, scrotal circumference, leg movement, head distance, and escape response) [5]. Additionally, Stafford et al. (2002) showed that lidocaine injected into the scrotum virtually abolished the cortisol response to banded castration [12]. Lidocaine may be effectively delivered to trauma sites non-invasively, through simple topical application [11]. 

Motivated by the results available in the literature, we have formulated castration bands containing therapeutic amounts of lidocaine, to deliver local pain relief to calves during banded castration. Based on industry stakeholder inputs, three different sized bands were developed for evaluation and field trialing in staggered parallel development streams to optimize resources and development timelines. The size categories include large (Callicrate^TM^) bands for bulls > 180 Kg, medium (beige bands) for bulls between 90 and 180 Kg, and small (green bands) for calves < 90 Kg. The band material was made of latex, which is sufficiently permeable to permit the release of local anesthetic from the lumen of the band to exterior surfaces, where it can then penetrate the scrotal tissue.

The objective of this study was to evaluate the lidocaine content, tissue transfer, and release kinetics of these lidocaine-loaded bands in vitro. Furthermore, the lidocaine delivery and pain mitigation effects of these lidocaine castration bands were determined in vivo through field trials. Manufacturing and release of lidocaine formulations for each device subtype was evaluated in a matrix-driven design. This development testing also evaluated occupational health and safety considerations, human factors, animal application factors, environmental durability, shelf life, and simulated use case stresses.

## 2. Materials and Methods

Three types of latex elastrator band—small (green), medium (beige), and large (Callicrate^TM^)—were loaded with free-base lidocaine (no epinephrine) according to a proprietary process that impregnates lidocaine and a skin permeation enhancer into the material of the band using a solvent carrier to produce LLBs, which were evaluated in this study. The first LLB was developed for small (4.3 cm circumference, 0.5 g) green elastrator bands recommended to be used in calves less than 90 Kg. These small bands had an average of 100 mg of lidocaine per band. The second LLB for the larger beige (5.3 cm circumference, 1.4 g) Tri-bander bands contained an average of 250 mg of lidocaine per band. The Callicrate^TM^ band, recommended for use in animals 180 Kg or greater, is much larger and forms a loop, so only the lower portion of the loop that would be in contact with tissue once applied was loaded with lidocaine. This lidocaine formulation had a blue dye added to help determine where the formulation was placed on the band and had an average of 86.4 mg of lidocaine per gram of band material. Under the conditions of our pilot manufacturing, this was found to be the maximum amount of lidocaine the bands were able to consistently retain. 

### 2.1. In Vitro Measurement of Initial Lidocaine Content in the Bands

The lidocaine from several castration bands was extracted in 100 mL tetrahydrofuran (THF) for 2 h at 37 °C. The extraction solution was diluted 12.5 times with acetone (in duplicate) and filtered through a 0.2 µm syringe filter prior to injection (in duplicate) onto a gas chromatography–mass spectrometry (GC–MS) instrument (see Table 1 for specifications and run conditions). A standard curve was obtained using solutions of 0.05, 0.1, 0.2, 0.4, 0.8 mg/mL of lidocaine prepared in the same solvent composition as the samples. The standards were injected (in duplicate) onto the same instrument, and the resulting data were used for quantitation.

### 2.2. In Vitro Measurement of Lidocaine Transfer into Steak Tissue over Time

The transfer kinetics of lidocaine into steak tissue (shown schematically in Figure 2) was determined for the small green bands as follows: A bovine sterling silver inside round steak was cut into 1 cm^3^ portions, and a LLB was placed around the steak and incubated for 0, 0.5, 5, 24, and 48 h at 37 °C, 5% CO_2_ (in triplicate for each time point). The LLB was removed from the steak cube following the prescribed incubation time, and both band and steak were stored in separate sealed containers at 4 °C. Steak cubes were cut into 10 uniformly sized pieces and extracted (twice) with 100 mL THF for 2 h at 37 °C. The extraction solution was diluted 2 times with acetone and filtered through a 0.2 µm syringe filter prior to injection (in duplicate) onto a GC–MS instrument. The LLBs from each time point were also extracted (three times) in 100 mL THF, and the extraction solution was diluted and analyzed as described above. 

### 2.3. Field Study Overview

The protocols were reviewed and approved by Chinook Contract Research’s Institutional Animal Care and Use Committee (OLAW #F19-00433, Application #14026-002-2,3,7) and the animals were cared for in accordance with Canadian Council on Animal Care (CACC) guidelines [13]. We developed and conducted a series of field studies to evaluate lidocaine release kinetics and pain mitigation effects in vivo for each band type. The three studies reported here first examined the green bands in May 2018, then the beige bands in July 2018, followed by the Callicrate^TM^ bands in October 2018. Each study was conducted at the same commercial veal operation and feedlot in southern Alberta. For each study, animals were weighed, ranked by weight, and allocated to treatment (LLB) or control (lidocaine injection) groups using random numbers. The animal was the experimental unit. All calves were intact male Holstein calves which were brought to the facility from multiple dairy farms in Southern Alberta. 

For study number 1 (green band) or 2 (beige band): Forty-eight calves (each study), weaned at birth and 3–4 weeks of age (53 ± 4.0 kg of body weight for study 1, 56.0 ± 8.0 kg of body weight for study 2), were used in an 8 day trial. Upon arrival at the farm (at approximately 4–5 days of age), calves were placed into individual pens (2.1 m × 0.76 m) with a partially slatted floor for waste removal. All animals were housed in the same room of a barn with forced air ventilation. Calves were vaccinated with an 8-way clostridial vaccine (Tasvax 8, Merck Animal Health, Kirkland, QC, Canada), respiratory disease vaccine (Inforce 3, Intranasal vaccine Zoetis Canada Inc., Kirkland, QC, Canada), topical parasiticide (Solmectin Pour-On for Cattle, Alberta Veterinary Laboratories Ltd., Calgary, AB, Canada), and allowed a minimum of two week adaptation period prior to the start of the trial. Calves were fed twice daily (morning and afternoon) via 7.6 L buckets attached to the front gate of each calf’s individual pen. One bucket contained milk replacer, 5 L per day (Mapleview Express Start + Deccox, Mapleview Agri Ltd., Drayton, ON, Canada) and the second bucket a dry pelleted complete feed (21% Protein Calf Starter, Landmark Feeds, Strathmore, AB, Canada). On the days of castration, biopsy, and biometrics collection, calves were manually restrained within their individual pen by a staff member utilizing proper restraining methods for young calves.

For study number 3 (Callicrate^TM^ band): Thirty calves, weaned at birth and 5 months of age (225 ± 20.0 kg of body weight), were used in an 8 day trial. Calves were comingled in a rectangular open-air pen (37 m × 55 m) with a wind break fence, a feed bunk along the front of the pen, straw bedding, and ad libitum well water supplied via an automatic watering system. Calves were vaccinated with an 8-way clostridial vaccine (Tasvax, Merck Animal Health, Kirkland, QC, Canada), respiratory disease vaccine (Inforce 3, Intranasal vaccine Zoetis Canada Inc., Kirkland, QC, Canada), and topical parasiticide (Solmectin Pour-On for Cattle, Alberta Veterinary Laboratories Ltd., Calgary, AB, Canada) at 1–2 weeks of age, and an 8-way clostridial vaccine again at castration. Calves also received an implant at Castration (Component E-S, Elanco Animal Health, Guelph, ON, Canada). Calves were adapted on farm for a minimum of four months prior to the start of the trial. Calves were fed once daily, approximately 3.5% of their body weight: 7.7–8.2 kg total mixed ration (TMR) comprising 30% barley silage, 15% corn dried distillers grain, 2.5% Premix containing mineral and vitamins (Landmark Feeds, Strathmore, AB, Canada), and 52.5% barley grain. On the days of castration, biopsy, and biometrics collection, calves were restrained in a manual squeeze chute (Lakeland, Stonewall, MB, Canada).

Animals in each study were randomly assigned to either experimental group 1 (control: lidocaine injection and castration with plain castration band) or 2 (treatment: castration using lidocaine-loaded band). Each experimental group contained 5 sampling time points (2, 6, 24, 48, and 168 h post-elastration), yielding 3 animals per treatment per time point (*n* = 3 per treatment per time point). A 20 mg/mL solution of lidocaine hydrochloride with 0.01 mg/mL epinephrine [5] (LIDO-2, Rafter 8 products, Calgary, AB, Canada; 1 mL for the Green- or Beige-banded small calves, 3 mL for the Callicrate^TM^-banded, larger calves) was injected subcutaneously into each of the spermatic cords in the neck of the scrotum of control animals before application of the plain castration band. LLBs and plain castration bands were administered to the treatment and control groups, respectively, at t = 0 h. All of the animals were weighed individually using a Jorvet scale (model J825PM, Jorgensen Laboratories, CO, USA) on day 0 and day 7 to calculate an average daily gain (ADG) for each experimental condition. 

### 2.4. Measuring Lidocaine Release Kinetics and Pain-Associated Behaviour In Vivo

The amount of lidocaine was determined from two tissue biopsies taken at each time point (2, 6, 24, 48, and 168 h after band placement) to determine in vivo lidocaine release rates. After removing the bands, two tissue biopsy samples (punch biopsy comprising both skin and subcutaneous tissue, 4 mm) were collected from the area that was in direct contact with the band. Each biopsy sample was placed in a microtube and stored at −80 °C. Tissue lidocaine extraction was initiated by chemically homogenizing the skin and subcutaneous tissue with 1 M Sodium Hydroxide and then neutralizing the dissolved solution with hydrochloric acid. A liquid–liquid extraction was performed with ethyl ether to separate and allow the removal of the organic layer. The remaining ether-lidocaine solution was evaporated to dryness and the residue re-dissolved with 1 mL of high-performance liquid chromatography (HPLC)-grade methyl alcohol. Lidocaine concentrations were measured by HPLC to determine lidocaine release (see Table 2 for specifications and run conditions).

At each time point, prior to punch biopsies, the animals were monitored for inflammation and pain-associated behaviour as follows: 

#### 2.4.1. Infrared Imaging of Scrotal Temperature

Images of the scrotal neck were captured above and below the placement of the elastration band using a thermal imaging camera (Model E75 1.1, lense 42°; FLIR Systems, Made in Estonia and Distributed by ITM Instruments Inc., Calgary, AB, Canada) from a distance of approximately 1 m (Emissivity = 0.95). The temperatures were registered and recorded at a single point (indicated by cross hairs in the thermographs) of the target scrotal areas approximately 2 cm above or below the castration band for each band type. Temperatures were also measured for the under belly of the animal, with the center of the scrotum targeted (beige and Callicrate^TM^ bands only; under belly temperatures were only considered and introduced at the conclusion of the green band field trial). Note also that the timing of the temperature measurements was updated between the green band trial (1, 1.5, 24, 48, and 144 h) and the later field trials (2, 6, 24, 48, and 168 h). 

#### 2.4.2. Electrostimulation

After infrared imaging, cutaneous stimulation was performed according to the procedure of Fierheller et al. [14] for the beige and Callicrate^TM^ banded experimental groups. This procedure was not performed for the animals castrated with green bands, as the evaluation methodology was not developed at the time of the green band field trial. A peripheral variable output nerve stimulator with the provided extension lead wires (one positive and one negative) and infant monitoring electrodes (8-1053-60; SunStim Peripheral Nerve Stimulator; Distributed by SunMed, Largo, FL, USA) was used to stimulate the skin over the banded area of the calf’s scrotum (after removing the band). Measurements were made at t = 2, 6, 24, 48, 168 h post-elastration. The uniformly effective stimulus setting was established to be 90 mAmp. The response of the animals to electrostimulation was graded [14] by a blinded large-animal veterinarian, using the following scale: 0 = no reaction, 1 = slight reaction (e.g., muscle tightening, twitch in the hind area, stiffening of stance), 2 = moderate reaction (e.g., flinch, slight jump with hind, tightening and stiffer stance), and 3 = severe reaction (e.g., severe flinch, obvious jump, abrupt stiffening). Reactions (if any) were immediate. 

### 2.5. Statistical Analyses

Statistical significance was determined using an unpaired, two-tailed *t*-test. Data were assessed for normality using a Shapiro–Wilk test. The cutoff for significance was *p* < 0.05. The experimental unit was defined as each individual animal. Statistical analyses were carried out in Prism v 8.4.3 (GraphPad Software, San Diego, CA, USA). 

## 3. Results

### 3.1. Transfer Kinetics for Lidocaine-Loaded Castration Bands (LLBs) In Vitro

GC–MS was employed to measure lidocaine concentrations in vitro (shown schematically in Figure 2). The amount of lidocaine in the LLBs was quantified by comparison to a standard curve (Figure 3A), indicating an initial content of approximately 80–210 mg of lidocaine per band (Figure 3B) for the three different LLBs developed. As an initial proof of concept, the transfer kinetics of lidocaine into ex vivo tissue (steak) was determined for the small green castration bands (Figure 4). The amount of lidocaine released from the bands was rapid for the first 30 min, followed by a slower release lasting at least 48 h (Figure 4A). Similarly, the amount of lidocaine transferred into the tissue was initially rapid, reaching approximately 7 mg in the first 30 min before plateauing at approximately 13.5 mg at 5 h, which was sustained for at least 48 h after band placement (Figure 4B). Taken together, these data suggest that LLBs are capable of delivering significant concentrations of lidocaine into tissue for at least 48 h.

### 3.2. LLBs Deliver Therapeutic Quantities of Lidocaine into Scrotal Tissue In Vivo

In order to confirm the efficacy of LLBs for lidocaine delivery into scrotal tissue during band castration, three field studies were performed to evaluate lidocaine release kinetics and pain mitigation effects in vivo. For each of the three band types (green, beige, and Callicrate^TM^), animals were castrated with either LLBs or with plain bands; the latter (control) group were injected with lidocaine immediately prior to band application for pain mitigation (in compliance with Institutional Animal Use and Care Committee recommendations). Scrotal tissue from the banding site was sampled, and the concentration of lidocaine therein was measured by HPLC over the course of a one-week study period (Figure 5). For the green bands (small calf bands for bulls < 90 Kg), LLBs generally delivered quantities of lidocaine that were similar to those of the lidocaine-injected control samples, although the 24 h tissue sample contained significantly more lidocaine in the LLB group than in the injected group (Figure 5A). In the case of the beige bands (medium bands for bulls < 180 Kg), LLBs generally delivered higher lidocaine concentrations relative to the injected controls, with statistically significant increases at 6, 24, and 168 h (Figure 5B). Similarly, LLB-treated tissue samples generally contained more lidocaine than their injected counterparts for Callicrate^TM^ bands (large bands for bulls > 180 Kg), with significant differences at all but the 2 and 24 h time points (Figure 5C). The concentrations of lidocaine in the LLB-treated tissue samples were generally in the range of 0.5–3.5 mg of lidocaine per gram of tissue and were generally highest after 6 h (Figure 5). Taken together, the results indicate that, over the short term, LLBs deliver at least as much lidocaine into scrotal tissue as injections into the spermatic cords. Over the long term (i.e., upwards of 6 h), LLBs can deliver significantly higher levels of lidocaine than injections.

As an indicator of overall performance, the animals were weighed on induction to this study (day 0) and on day 7 to calculate an average daily gain (ADG) for each experimental condition. ADG was comparable for LLB-treated versus lidocaine-injected animals, indicating similar performance between these groups (Table 3). No statistically significant differences in ADG were observed between the LLB-treated and injected animals (statistical significance was determined using an unpaired, two-tailed *t*-Test).

Infrared thermography is a non-invasive means of measuring tissue inflammation [15,16]. Accordingly, we used thermal imaging to assess the temperature of different scrotal regions, and representative images are presented in Figure 6, with the results quantitated in Figure 7. As expected, scrotal temperatures were generally lower below the banding site over time, consistent with the restriction of blood flow imposed by the bands (Figure 7). No significant differences in scrotal temperature were observed for LLBs relative to their injected counterparts, indicating similar levels of inflammation (Figure 7). However, at 168 h, the under-belly temperature was significantly lower for Callicrate^TM^ LLBs relative to their injected controls, suggesting decreased inflammation for the LLB-treated animals at this time point (Figure 7C). The number of tail flicks and average heart rate were also measured, and indicated no significant differences between LLB-treated and lidocaine-injected animals (Appendix A). 

Finally, as a measure of scrotal numbness at the castration sites, electrostimulation was used to elicit a pain response, which was graded according to the scale detailed in Materials and Methods. For both the beige and Callicrate^TM^ bands, LLBs gave comparable pain responses to their injected controls up to 48 h (Figure 8). However, at 168 h, LLBs gave significantly decreased pain responses relative to the injected controls, indicating that LLBs are more effective at inducing long-term numbness in the relevant tissue (Figure 8). 

## 4. Discussion

The main goal of this study was to establish whether lidocaine could be topically delivered from elastration bands into tissue for pain mitigation during banded castration. As an initial proof of concept, the transfer kinetics of lidocaine into ex vivo tissue (steak) was successfully determined. These data suggest that lidocaine delivery by LLBs into tissue is initially fast, followed by a sustained delivery lasting at least 48 h (Figure 4). Similar results were obtained in a study that evaluated lidocaine permeation across rabbit ear skin from a transdermal patch [17]. We note that the transfer kinetics of lidocaine into steak tissue was only determined for the small green bands; given the conserved latex composition of the three different band types and the similar amount of lidocaine in each band (Figure 3), it is reasonable to expect similar results for both the beige and Callicrate^TM^ bands. We acknowledge that steak and scrotal tissue have different characteristics that could affect lidocaine release and diffusion. Accordingly, we have moved future in vitro lidocaine transfer experiments into tissue models incorporating epidermal and dermal tissues to better replicate bovine scrotal tissue. However, at the inception of this study, these tissue models were not yet in place and therefore steak tissue was used to provide a rough approximation of lidocaine transfer kinetics prior to field studies.

The in vitro observations were generally corroborated by the in vivo data (in which all three band types were assessed): tissue concentrations of lidocaine were similar over the short term (i.e., up to 6 h), but higher over the longer term (i.e., up to 7 days) for LLBs relative to injected control animals (Figure 5). These results confirm that lidocaine absorption is initially similar for the two routes of administration (i.e., transdermal vs subcutaneous injection), despite the difference between the two tissue types. However, lidocaine tends to be significantly higher in LLB-treated vs injected samples starting at approximately the 6 h time point, likely reflecting the fact that lidocaine injection is a one-off treatment at Time 0, while LLBs were shown continue to deliver lidocaine for at least 7 days. This would have obvious benefits for both acute and chronic pain management during castration, with immediate pain relief at the time of band application and sustained pain-relief over the course of at least 7 days. In order to confirm whether LLBs continue to deliver therapeutic quantities of lidocaine beyond 7 days, it will be necessary to conduct a longer-term study. 

The concentrations of lidocaine in the LLB-treated tissue samples were generally in the range of 0.15–3.5 mg of lidocaine per gram of tissue. In contrast, lidocaine levels were generally lower for injected tissue samples—in fact, beyond the 6 h time point, they were generally either undetectable or around 0.1 mg of lidocaine per gram of tissue (100 µg/g of tissue). For reference, tissue concentrations below 100 µg/g are not considered therapeutic [18]. It thus appears that lidocaine injection is often—but not always—sufficient to achieve therapeutic levels of lidocaine in the scrotal tissue, while LLBs consistently yield levels well above this threshold. Accordingly, with regard to the performance metrics evaluated in this study, LLBs often perform similarly to lidocaine injection. For instance, LLBs yielded no statistically significant differences in ADG versus lidocaine injection (Table 3), suggesting that LLB-treated animals gained at least as much weight as the lidocaine-injected animals. ADG is a common indicator for post-operative performance in livestock [19,20,21]. The lack of statistically significant differences in ADG could also reflect the high variability in this data due, at least in part, to the low *n* value (especially for the green bands). The green-banded ADG sampling is especially low (*n* = 3) because measuring ADG was only considered and introduced at the conclusion of the green band trial. Based on this pilot work, it would be interesting to measure ADG in a future trial involving a larger number of animals. Additionally, scrotal and underbelly temperatures were generally similar for LLB-treated versus injected animals according to infrared thermography (Figure 6 and Figure 7), with only one statistically significant difference: for Callicrate™ bands, at 168 h, the under-belly temperature was significantly lower for LLB-treated vs injected animals (Figure 7C). In this context, lower temperature indicates reduced blood flow and/or decreased inflammation [15,16]. This difference might reflect the fact that lidocaine levels were significantly higher in the LLB-treated tissues at this time point; in fact, lidocaine was undetectable in the injected samples (Figure 5C). Finally, LLB-treated animals had similar responses to electrostimulation relative to lidocaine-injected animals (Figure 8), with significantly reduced responses only at the 168 h time points of the Callicrate™ and Beige-banded animals. This might reflect the significantly reduced levels of lidocaine in the injected animals at this time point (Figure 5B,C), even though the injected levels were above the therapeutic threshold for the Beige-banded cohort (Figure 5B). Moreover, the average heart rate and number of tail flicks were not significantly different between LLB-treated and lidocaine-injected animals (Appendix A), suggesting a similar performance of these two methods of pain mitigation. From a safety perspective, no adverse events or mortalities were observed in either the treatment or control groups over the course of the in vivo field studies. Taken together, these results indicate that LLBs perform at least as well as lidocaine injections for short-term lidocaine delivery and pain mitigation (up to approximately 6 h). Moreover, LLBs generally outperform lidocaine injections for long-term delivery and pain mitigation (up to at least 7 days). 

Of note, we elected to compare the performance of the LLBs to injections of lidocaine into the spermatic cord rather than a ring block delivery because the LLB technology incorporates a skin permeation enhancer which is designed to deliver anesthetic deep into the scrotal tissue. Therefore, the cord injection was chosen over the ring block to facilitate a more accurate comparison to the LLB delivery profile. We acknowledge that future studies could also include a ring block control group to make an evaluation of deep versus surface anesthesia. 

The relatively short duration of lidocaine delivery and pain mitigation reported here for lidocaine injection at the time of castration is in keeping with other published studies. For instance, in a study of the effect of lidocaine on indicators of pain and distress during and after knife castration in beef calves, Melendez et al. (2018) noted that salivary cortisol concentrations were only significantly lower in lidocaine-injected versus untreated animals at 0.5 and 1 h after castration [5]. Stafford et al. (2002) found that lidocaine injection significantly reduced mean plasma cortisol concentrations in calves in response to band castration over a similar time period (0.5 to 3 h post-banding) [12].

Generally, this pilot work facilitated the prototype development and field validation of lidocaine-loaded castration bands for bovine pain mitigation. Due to the complex, multi-faceted nature of accurately assessing and reporting pain in animal studies [3,4], a robust, unambiguous measurement protocol should be employed. To evaluate the effectiveness of the LLB compared to lidocaine injection, the methods presented here combined the measurement of two behavioural responses (tail flicks and response to electrocutaneous stimulation) with the quantitation of one performance indicator (ADG), two physiological indicators (tissue inflammation and average heart rate), and tissue lidocaine levels. In future studies, it might be advantageous to include additional physiological indicators of pain and inflammation, such as substance P and salivary cortisol levels. While the field studies presented here lasted 7 days, future work should increase the study duration to approximately 3–6 weeks to reflect the actual duration of banded castration. 

A secondary objective of this work was to develop an objective assessment tool that generated more consistent results between different observers and different instruments [19] for the specific application of evaluating topical lidocaine delivery to generate an effective local anesthetic effect. To this aim, we took cues from the work of Melendez et al. (2018) [5], Musk et al. (2017) [19], and Winder et al. (2017) [22], and developed an evaluation protocol that incorporated a combination of physiological and behavioural metrics to evaluate pain response over the course of our field trials. Moreover, we adopted the use of an electrocutaneous nerve stimulation protocol (a standard of practice in guiding and monitoring peripheral nerve blocks) [23,24] to evaluate the loss of sensation in the animals’ scrotums over acute and chronic observation periods. For the purposes of this study, we define ‘acute’ and ‘chronic’ as the first several hours (up to approximately 6 h) and the first several days (up to at least one week) post-castration, respectively. Our method also incorporated the use of precision analytical HPLC to detect and quantify the lidocaine directly in the sampled tissue of interest. This analysis was originally incorporated into the trial design to better understand the LLB device performance; however, the addition of this analytical step provided this protocol with the means to attain comparable objective measurements between instruments (electrocutaneous stimulation results assessed against physiologically relevant levels of lidocaine in the tissue). 

The ultimate goal of this work is to provide producers with a practical and effective tool to provide long-term pain control for banded castration. The benefits to the industry include a simplified banding procedure (eliminating the need for lidocaine injection), reduced costs, societal and animal welfare benefits associated with improved pain control, and a potentially improved castration device (i.e., through increased ADG and enhanced recovery). Finally, as pain management is important for the well-being of the animal and castration recovery time [3,4], this work has obvious benefits for animal welfare. 

## 5. Conclusions

Lidocaine-loaded elastration bands are capable of delivering significant quantities of lidocaine into scrotal tissues over at least seven days. This approach could provide long-term pain mitigation to the animals and, by avoiding surgery or the administration of injections, decrease time and handling costs for the producer. 

## Figures and Tables

**Figure 1 animals-10-02363-f001:**
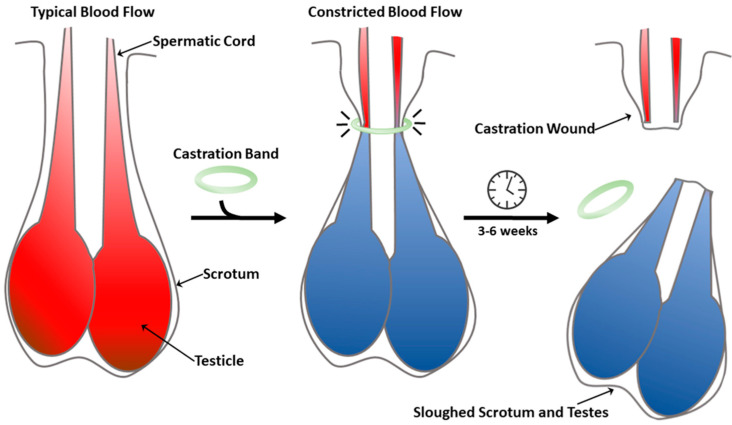
Castration procedure using elastic castration bands.

**Figure 2 animals-10-02363-f002:**
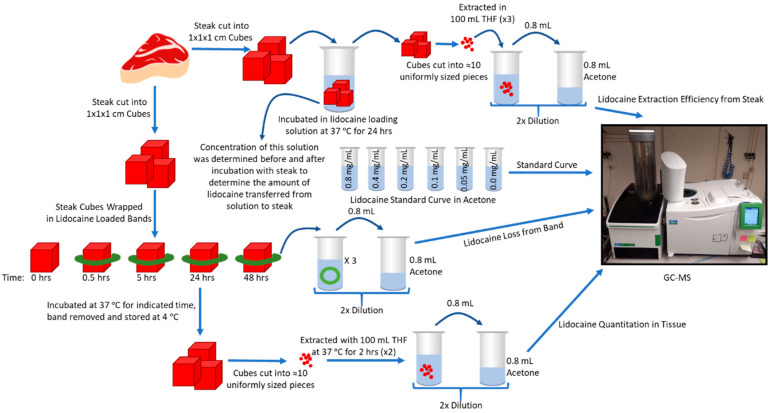
Schematic overview of lidocaine quantitation in steak tissue.

**Figure 3 animals-10-02363-f003:**
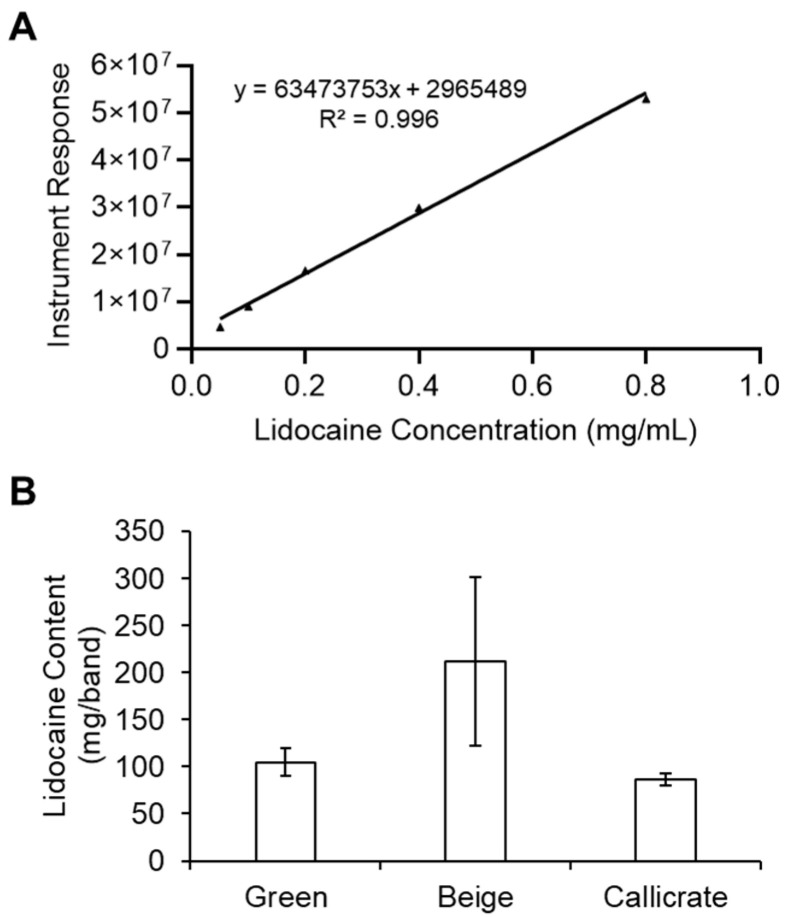
Calibration curve and lidocaine quantitation by GC–MS. (**A**) Standard curve for various lidocaine concentrations quantified (in duplicate) by GC–MS. (**B**) Quantitation of lidocaine extracted from each type of lidocaine-loaded castration band (LLB). Note that, for the Callicrate bands, only a section of the band (corresponding with the surface that would contact the animal’s scrotum) was analyzed. Bars represent the mean ± standard deviation for 6 (green), 3 (beige), or 4 (Callicrate) independent replicates.

**Figure 4 animals-10-02363-f004:**
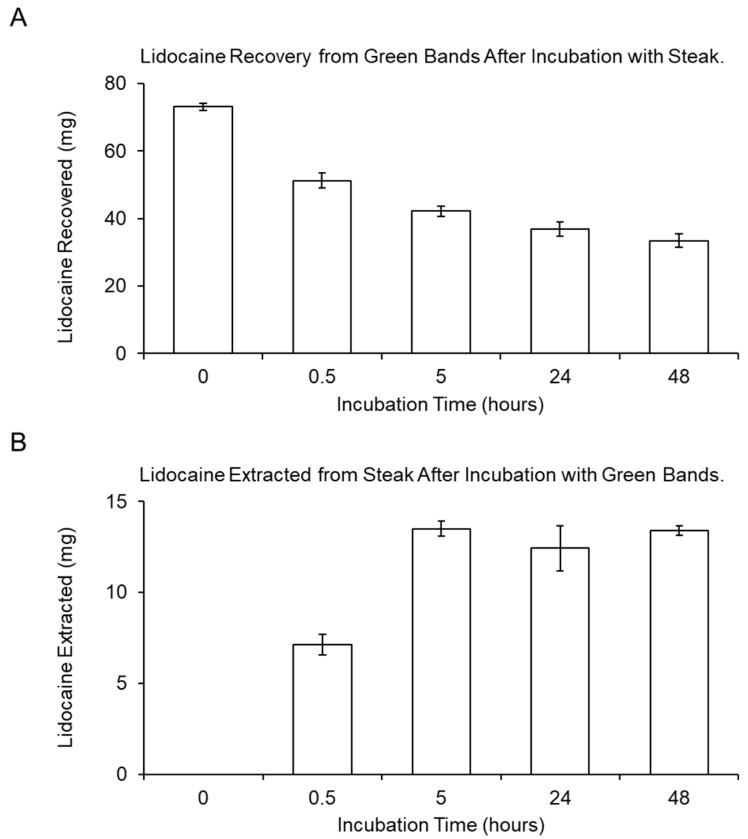
Lidocaine transfer kinetics between lidocaine-loaded, green castration bands and steak tissue. (**A**) The amount of lidocaine recovered from lidocaine-loaded green castration bands (LLBs) following various periods of incubation wrapped around steak tissue. (**B**) The amount of lidocaine extracted from steak tissue following various incubation periods wrapped with LLBs. Lidocaine was quantified by GC–MS. Bars represent the mean ± standard deviation for three independent replicates.

**Figure 5 animals-10-02363-f005:**
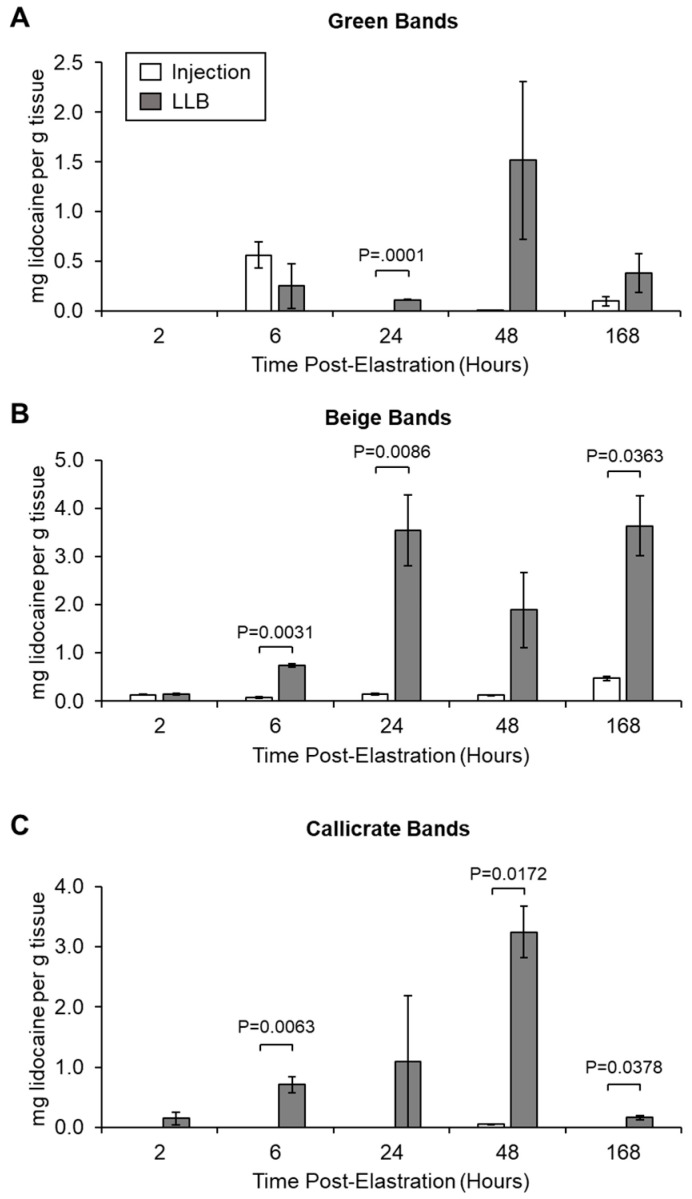
Lidocaine concentrations from scrotal biopsy samples collected over time. Time indicates the duration the band was on the animal prior to biopsy. Biopsies from animals treated with green (**A**), beige (**B**), or Callicrate^TM^ (**C**) bands (LLB), or their corresponding controls (lidocaine injection), were homogenized, extracted, and the lidocaine content measured by HPLC. Bars represent the mean ± standard error of the mean (SEM) for three independent replicates per time point.

**Figure 6 animals-10-02363-f006:**
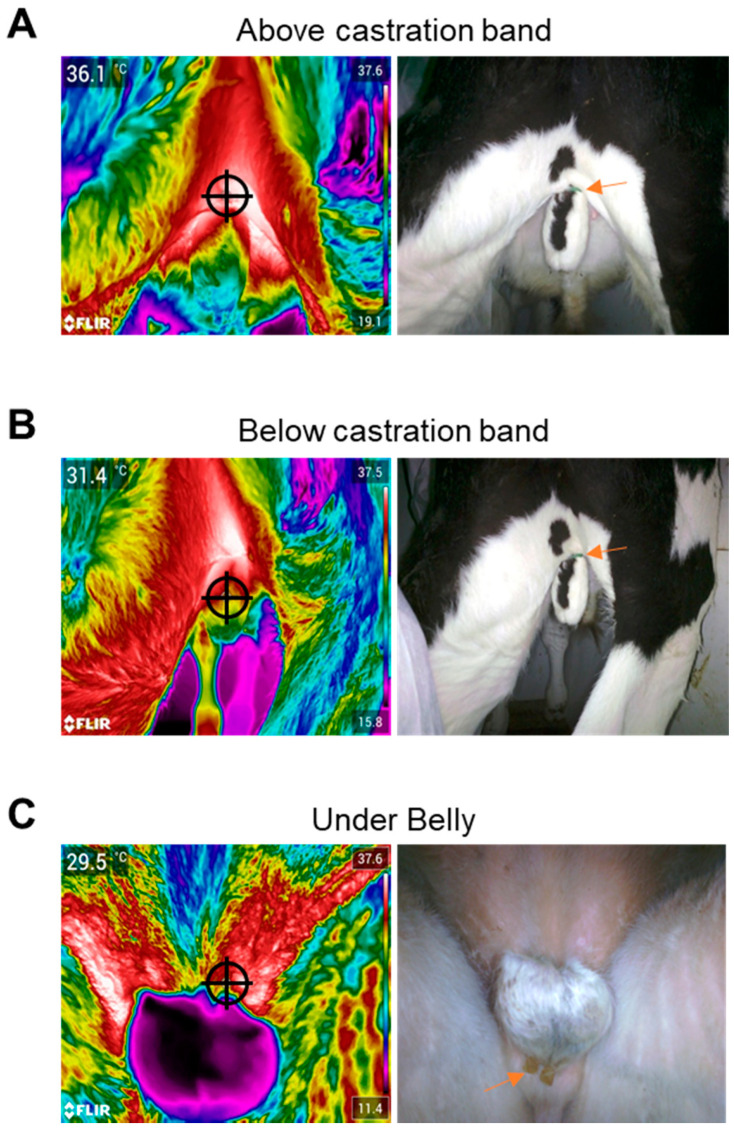
Infrared imaging of animals with castration bands. Representative infrared images (left) and corresponding photographs (right) are presented for (**A**) above the castration band, (**B**) below the band, and (**C**) the underbelly of the animal. Cross hairs (left-most images) indicate the point at which the temperature was registered. Orange arrows (right-most images) indicate the bands. The temperatures were quantitated and are presented in Figure 7.

**Figure 7 animals-10-02363-f007:**
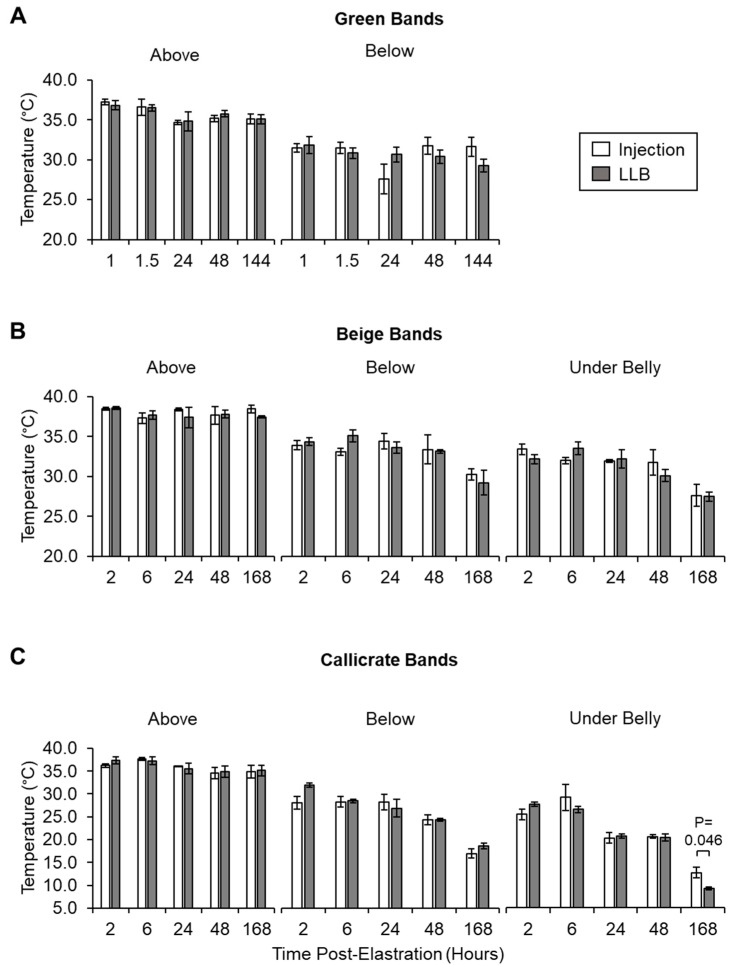
Scrotal temperatures of control and treatment calves above and below castration band placement over time. Scrotal temperature above and below the green (**A**), beige (**B**), and Callicrate^TM^ (**C**) castration bands were plotted against the duration of band placement for test (LLB) and control (injection) animals. The temperature of the animal’s belly was also measured for the animals treated with beige (**B**) and Callicrate^TM^ (**C**) bands. Bars represent the mean ± standard error of the mean (SEM) for three independent replicates per time point. Statistical significance was determined using an unpaired, two-tailed *t*-test. Data were assessed for normality using a Shapiro–Wilk test. The cutoff for significance was *p* < 0.05.

**Figure 8 animals-10-02363-f008:**
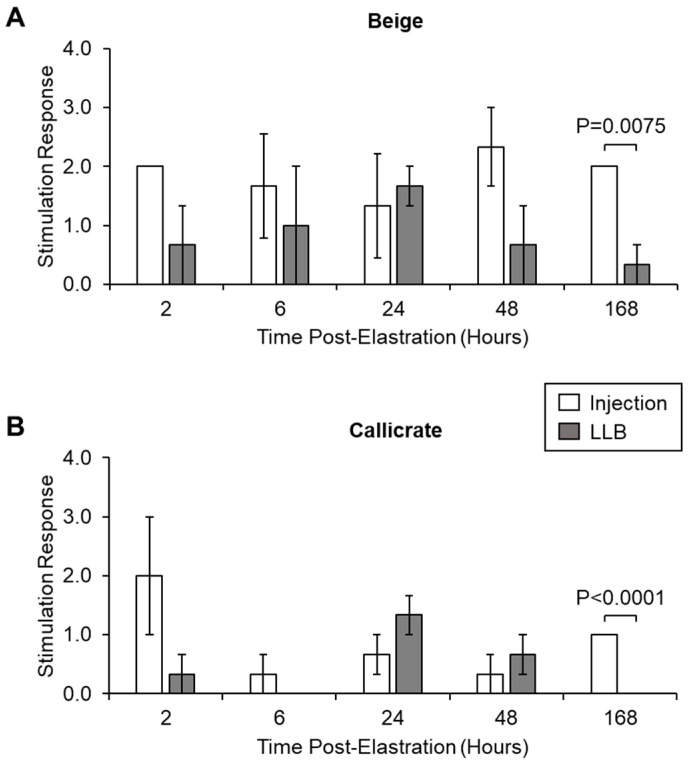
Response of control and treatment calves to electrostimulation at the castration site. Animals were treated with electrostimulation at the banding site and their pain response measured according to a graded scale as described in Materials and Methods. Graded responses for the beige (**A**) and Callicrate^TM^ (**B**) bands were plotted against the duration of band placement for test (LLB) and control (injection) animals. Bars represent the mean ± standard error of the mean (SEM) for three independent replicates per time point. Statistical significance was determined using an unpaired, two-tailed *t*-test. Data were assessed for normality using a Shapiro–Wilk test. The cutoff for significance was *p* < 0.05.

**Table 1 animals-10-02363-t001:** GC–MS specifications and run conditions.

Instrument or Parameter	Details
GC–MS	PerkinElmer Clarus 680-AxION iQT
Column	Elite-5MS, 30 m × 0.25 mm, 0.25 µm
Carrier Gas	Nitrogen
Analysis Time	15 min
Flow Rate	1 mL/min
Injection Volume	0.5 µL
Injection Temperature	250 °C
Split Ratio	20:1
Oven Program	60 °C → 300 °C over 15 min
Detection	MS Scan 50—1045 *m*/*z*
GC–MS	PerkinElmer Clarus 680-AxION iQT
Column	Elite-5MS, 30 m × 0.25 mm, 0.25 µm

**Table 2 animals-10-02363-t002:** HPLC specifications and run conditions.

Instrument or Parameter	Details
HPLC	Hewlett Packard 1100 Series
Column	Kinetex 2.6 µm C18 (100A, 150 × 4.6 mm)
Mobile Phase	40:60 (Acetonitrile: 0.05 M Sodium Phosphate Buffer Solution), 0.05% Diethylamine
Analysis Time	6 min
Flow Rate	1 mL/min
Injection Volume	10 µL
Column Temperature	28 °C
Detector	Variable Wavelength Detector (VWD)
Wavelength	210 nm
Bandwidth	10 nm

**Table 3 animals-10-02363-t003:** Average daily gain for control versus treatment animals.

Average Animal Size at Banding	Band Type	Treatment Group	ADG ^a^ (Kg/Day)	± SD ^b^
225 ± 20 Kg	Callicrate^TM^	Injection (*n* = 14)	0.43	0.52
		LLB (*n* = 15)	0.80	0.67
56 ± 8 Kg	Beige	Injection (*n* = 21)	0.93	0.33
		LLB (*n* = 20)	0.99	0.23
53 ± 4 Kg	Green	Injection (*n* = 3)	0.73	0.27
		LLB (*n* = 3)	0.89	0.65

^a^ ADG: average daily gain, calculated 7 days post-castration. ^b^ SD: standard deviation.

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
