# Peer review of "Development and Field Validation of Lidocaine-Loaded Castration Bands for Bovine Pain Mitigation"

_animals, 2020, doi:10.3390/ani10122363_

Round 1
Reviewer 1 Report
The discussion of the paper has greatly improved.
A minor change:
Add a '±' symbol prior to the SD of the calves weight. Add a decimal point to the SD of the weights e.g. 53 ± 4.0
Author Response
The discussion of the paper has greatly improved. Thank you very much your first round of revisions. Your suggestions greatly helped improve the paper.
A minor change:
- Add a '±' symbol prior to the SD of the calves weight. Add a decimal point to the SD of the weights e.g. 53 ± 4.0 Lines 159, 174
Reviewer 2 Report
- Great idea to incorporate Lidocaine into the bands used for castration. Long overdue. It will have profound implications for how we manage calves because it will become the band of choice for humane concerns.
- Great description of how to assess levels of Lidocaine in the tissues.
- My only concern is comparing the insensitivity produced by an injection into the spermatic cord to that of an insensitivity produced by a effective diffusion into the skin at the scrotal neck. I always noted that if I ring block the scrotum, the bull can't feel the Newberry knife opening the scrotum, but seriously reacts when I emasculate the testicles and vice versa.
- The predominant pain created by banding is from the scrotal skin and blocking the spermatic cord would not stop that pain.
- Note: This difference in anatomic innervation should be explained in your discussion.
Author Response
- Great idea to incorporate Lidocaine into the bands used for castration. Long overdue. It will have profound implications for how we manage calves because it will become the band of choice for humane concerns. Thank you very much. We are working hard to get this to market to support producers.
- Great description of how to assess levels of Lidocaine in the tissues. Thank you very much. We are working hard with other collaborators on future studies to improve on this and working with VDD to get this product approved with a pain mitigation label claim.
- My only concern is comparing the insensitivity produced by an injection into the spermatic cord to that of an insensitivity produced by a effective diffusion into the skin at the scrotal neck. I always noted that if I ring block the scrotum, the bull can't feel the Newberry knife opening the scrotum, but seriously reacts when I emasculate the testicles and vice versa. Excellent point. We now acknowledge this in the discussion.
- The predominant pain created by banding is from the scrotal skin and blocking the spermatic cord would not stop that pain. Excellent point. We now acknowledge this in the discussion.
- Note: This difference in anatomic innervation should be explained in your discussion. Thank you for this excellent comment. We have added the following comment to the discussion: “Of note, we elected to compare the performance of the LLBs to injections of lidocaine into the spermatic cord rather than a ring block delivery because the LLB technology incorporates a skin permeation enhancer which is designed to deliver anesthetic deep into the scrotal tissue. Therefore, the cord injection was chosen over the ring block to facilitate a more accurate comparison to the LLB delivery profile. We acknowledge that future studies could also include a ring block control group to make an evaluation of deep versus surface anesthesia”.
Reviewer 3 Report
Line 31- by slower you mean a longer time for scrotum to slough from animal? Current wording suggest the procedure takes longer.
Line 61- many producers routinely use meloxicam at all castration events (including spring calf processing) and I would consider this at the herd level.
Line 112 - a more detailed description of the how the lidocaine is loaded into the band without revealing any proprietary information would be helpful.
Line 150 - reference field studies.
Line 261 - Figure 3 should be on same page as graphs.
Line 306 - state statistical analysis used.
Table 3 - include number of animals in each treatment group, statistical analysis used.
Figure 6 - Small grey cross hairs difficult to see. Change color?
Figure 7 - state statistical analysis used.
Line 357 - elastration bands into tissue for pain....
In general a well conducted study on a practical technique an important field of study for bovine production.
Author Response
Line 31- by slower you mean a longer time for scrotum to slough from animal? Current wording suggest the procedure takes longer. Removed the phrase “Although slower than other approaches” to remove confusion.
Line 61- many producers routinely use meloxicam at all castration events (including spring calf processing) and I would consider this at the herd level. -Agreed; we were referring to injectable anesthesia. We have modified line 61 to now read: “Existing techniques (i.e. ring blocking the scrotum or spermatic cord injection) to deliver anesthesia during castration are not practical at the herd level.”
Line 112 - a more detailed description of the how the lidocaine is loaded into the band without revealing any proprietary information would be helpful. Agreed. At the time of the submission of this work, the patent has been applied for but not yet granted and we need to be considerate of protecting the IP associated with this. In order to balance this protection with consideration to the reader we respectfully offer this wording: “Three types of latex elastrator band—small (green), medium (beige), and large (CallicrateTM)—were loaded with free-base lidocaine (no epinephrine) according to a proprietary process that impregnates lidocaine and a skin permeation enhancer into the material of the band using a solvent carrier to produce LLBs, which were evaluated in this study”.
Line 150 - reference field studies. The studies were developed and executed by us. There are no reference studies for this work as far as we are aware. We describe the conduct of this work in this paper for the first time. We have offered additional wording to further clarify this in lines 151 and 152.
Line 261 - Figure 3 should be on same page as graphs. DONE.
Line 306 - state statistical analysis used. (see materials and methods section 2.5: Statistical significance was determined using an unpaired, two-tailed T-test. Data was assessed for normality using a Shapiro-Wilk test. The cutoff for significance was P < 0.05. The experimental unit was defined as each individual animal. Statistical analyses were carried out in Prism v 8.4.3 (GraphPad Software).
Table 3 - include number of animals in each treatment group, statistical analysis used. Number of animals listed by group in brackets. No statistical analysis was done outside of presenting the standard deviations. We ran a T-test and this yielded no statistically significant differences in ADG versus lidocaine injection.
Figure 6 - Small grey cross hairs difficult to see. Change color? DONE…thank you. Great comment.
Figure 7 - state statistical analysis used. (see materials and methods section 2.5: Statistical significance was determined using an unpaired, two-tailed T-test. Data was assessed for normality using a Shapiro-Wilk test. The cutoff for significance was P < 0.05. The experimental unit was defined as each individual animal. Statistical analyses were carried out in Prism v 8.4.3 (GraphPad Software). Also added to Figure 8 for completeness.
Line 357 - elastration bands into tissue for pain.... DONE…thank you. Great comment.
In general a well conducted study on a practical technique an important field of study for bovine production. Thank you very much. We are working hard to get this to market to support producers.
Round 2
Reviewer 2 Report
I have no further suggestions to make
This manuscript is a resubmission of an earlier submission. The following is a list of the peer review reports and author responses from that submission.
Round 1
Reviewer 1 Report
This manuscript has the objective to evaluate the lidocaine content, tissue transfer and release kinetics of lidocaine-loaded bands in vitro and the effects of pain mitigation in vivo. It is a novel approach that can benefit the industry and the well-being of the animals. However, there is a lack of information throughout the article. The material and methods section needs a better description and the experimental design seems very weak. As far I can understand from the material and methods the in vivo results are done with 3 animals per treatment, with the animal variability that exists, I have serious doubts that these results are representative. The statistical analysis should be improved. Also, the focus of the discussion should be addressed to the objectives of the study.
L64: add reference
L81 to 87: This paragraph seems out of place. It may fit better in line 98.
L112 to 120: What was the criterium for the dosage of the bands?
L132: Why the authors used a stake tissue when the scrotal tissue have different characteristics.
L134: Why only with the green bands. Do you expect the same results with the other type of bands?
L135: Check incubated time 5h as in the figure it is 6h.
L149: Reference number of the approval committee is missing.
L149: Each one of the studies should be properly described. Housing, diet, type of animals, etc.
L150: This sentence is confusing…what is the exception? Where the control animals administered with two different routes of administration?
L153: If one of the parameters collected was BW, please state the initial body weight for each one of the groups and age if possible.
L153: If studies were run in May, July and October, why in line 151 was reported that two trials were run in parallel?
L157: Please indicate the 8 sampling time points
L158: Please confirm that you collected data only for 3 animals per treatment per experiment. Do you think that is enough to have representative results? There were always the same animals that were evaluated?
L158: Reference the dosage of lidocaine used…I think most of the authors cited thereafter used different amounts.
L162: All animals were weighed or just the three animals that you collected data?
L164: The absorption of the lidocaine may be different when applied transdermal as with the LLBs or subcutaneously into each spermatid cord. The biopsy tissue described may not be representative of the lidocaine injected vs the transdermal.
L165: Did you do biopsies without any use of pain mitigation in CTR animals? Did you have in account the stress and pain of that procedure at the time of evaluate the parameters evaluated?
L166: Here do you report that you collected skin and tissue biopsy. Why you did perform the analysis separately? Why in the results section only appears the tissue and there are no results of the skin? Why you did not perform the kinetics in blood?
L168: Please specify if the area of the sample was above of below the band.
L175: The inflammation and pain-associated behaviour data was collected previous or after the biopsies? The days of biopy collection for the green bands do not correspond with the days of the other sample collections. Please confirm that it was like this, and describe it in the manuscript as it may have an effect on the results.
L180: Did you registered the temperature in one point as shown in Figure 6? How did you define that point? How do you know then is the maximum temperature of the area? Did you use maximum temperature of average temperature?
L180: Missing the manufacturer of the camera
L181: Please describe time points for each one of the studies as in figure 6A data is not collected at the same time points than B and C. Why there is no temperature underbelly for the green bands?
L183: Please indicate here that only was done for beige and Callicrate groups
L185: Reference and characteristics of the electrode are missing
L186: At what time points it was performed? For how long the reaction of the animal was observed?
L192: I don’t think this is the proper statistical analysis for the in vivo data, probably a mixed model with repeated measures would be more adequate unless the animals evaluated each time were different and therefore the results with only 3 animals are very compromised. Also the electro-stimulation data is categorical and therefore another type of analysis should be performed. Please describe if the data had a normal distribution, what was the experimental unit etc.
L194: The results section have a lot of information that should be moved to material and methods if not repeated. Also data should be reported with standard error of the mean not standard deviation that make the results difficult to follow with the big variation in the data.
L309: The discussion should be focus on the results of the study.
Reviewer 2 Report
This article describes an initial attempt to validate lidocaine loaded castration bands which is an innovative way to mitigate pain after band castration.
Overall, there is missing information throughout the manuscript. The experimental design of this study is confusing as in vitro and in vivo measurements were not collected from all groups at all times and the rationale behind this is not explained. Information in the manuscript is repeatedly misplaced between the results, materials and methods and the discussion sections. Vital information is omitted in the materials and methods section and in depth discussion of the results is lacking. In addition, the conclusions made are not supported by the discussion.
Line 64: add reference for maximal plasma concentrations of meloxicam.
Line 81: what does (3) mean?
Line 88: remove ‘s’ from pain.
Line 124: first time GC-MS abbreviation appears in the text. Add the name of abbreviation and place abbreviation in parenthesis.
Line 126: add ‘of’ between mg/ml and lidocaine.
Line 135: incubated time in the text is 5h while in Fig 2. 6h are reported.
Line 141: add ‘described’ between as and above.
Line 148: is there an ACC number?
Line 149: add CCAC reference.
Line 154: add average BW and breed of calves and housing conditions. Describe when were the 8 sampling time points collected? Where the sampling points done at the same time for both control and lidocaine calves? How were calves restrained at the time of sampling?
Line 157: sentence not clear, what does this mean? Is the n= 24 or n= 3 per sampling point?
Line 158: add in the text that the lidocaine used was lidocaine with epinephrine. Describe if the same or a different lidocaine was used to load the bands.
Line 158: how much lidocaine did the medium size calves receive?
Line 162: add make and model of scale used to weigh calves.
Line 165: five sampling points are described for biopsies but only two biopsies were collected. Clarify if two biopsies were collected at each time point.
Line 165: how were the animals restrained at the time of biopsy collection?
Line 173: replace ‘concentration was’ with ‘concentrations were’.
Line 174: describe the make and model of the HPLC and analysis conditions.
Line 180: add thermal camera manufacturer, distance and emissivity coefficient used.
Line 181: clarify what time points, line 157 says 8 time points and in line 165 only 5 time points are mentioned.
Line 184: add nerve stimulator manufacturer.
Line 185: add the make and model of electrode and the size. How many electrodes were used?
Line 186: unclear wording of sentence. Describe how this was established.
Line 188: describe how does tension in the scrotum looks like.
Line 190: at which sampling points was the electrostimulation done?
Line 190: describe the order of procedures during sampling.
Line 191: What statistical program was used for the statistical analysis? Does the data meet the t-test criteria such as scale of measurement, simple random sample, normal distribution, sample size and homogeneity of variance. If these assumptions were met, describe how there were met. It would be interesting to consider the relationship of the in vivo measures by including a repeated measures analysis, as sample are collected from the same animals at different time points.
Line 191: what was considered the experimental unit?
Line 196 to 199. Repetitive information that has already been mentioned in materials and methods. This section should only have results.
Line 201: why were the transfer kinetics limited to the green bands? It would be interesting to know if there was any difference in transfer kinetics between the different bands, specially due to the different lidocaine concentrations.
Line 226. Explain the rationale of using steak tissue instead of tissue that would resemble the neck of the scrotum that has skin.
Line 253: ADG is a performance parameter, not an indicator of overall stress and pain associated behaviours.
Line 257 to 258: same as before, there is an explanation as to why there were no differences between treatments. This should be mentioned in the discussion section not in the results section.
Table 2. the number of animals in the green group is extremely low.
Figure 5. discuss (in the discussion section) why there are no lidocaine levels measured for the callicrate band at 24h.
Line 256: Results section should only report results, not conclusions from the results. In addition the conclusion ‘indicating similar levels of pain and discomfort between these groups’ can’t be concluded based on ADG. Several castration studies have reported no differences in ADG between castrated and non castrated calves. This does not mean that animals experienced the same amount of pain.
Line 263 to 266: this sections should be moved to the materials and methods sections. These are not results. It says infrared was collected only from Callicrate calves but in Fig 7 there are results for all three band types. Clarify.
Line 266: it would be useful to show the reader the area that was used to measured scrotal temperature. If this is represented by the cursor, what guidelines did you use to select this location consistently with all animals. If an area was used this can be added to Fig 6. Were maximum or average temperatures used for analysis?
Fig 7. There is information in Fig 7 that should be described in materials and methods such as why there is a difference in the hours that the scrotal temperature was collected between band sizes and why wasn’t the under belly temperature collected for green bands.
Line 287-288 should not be included in this section.
Line 290-293: again, this should be in the materials and methods section.
Line 298-300: this does not belong to the results section.
Line 303-305: this sentence does not belong here.
Line 310: Was there a reason for not assessing lidocaine recovery and extraction at 168h when biopsies were being collected?
Line 311-313: this statement is incorrect as the in vitro data is limited to 48 h and the in vivo data for 7 days, while castration takes between 3 to 6 weeks.
Line 318: the authors should discuss why differences are observed for stimulation response and scrotal temperature at 168 h when biopsy concentrations are above the therapeutic level in both banded and injected calves.
Line 318: if trends are going to be discussed these should be described in the statistical analysis section.
Line 320: ADG is NOT a good indicator of pain.
Line 331-332: this is the first time that this has been mentioned. There should be a section in materials and methods describing these parameters and results for those parameters. If not, remove. What does ‘biochemical’ mean?
Line 335: replace ‘physiological’ for ‘behavioural’
Line 336: remove ‘accurate quantitation’
Line 336: the authors mention how a robust protocol should be used and then describe 3 parameters that they measured: ADG, scrotal temperature and electrostimulation. Limitations of the present study can be included in this section, acknowledging the limited number of parameters assessed (1 physiological, 1 behavioural and 1 performance).
Line 337-341: This sentence is confusing. Please reword. Are the authors suggesting that heat rate and cortisol are not good parameters due to handling? If they are not good parameters why where they evaluated? I do not agree with the recommendation of only using 3 parameters for future development or registration as this is a limitation of the present study.
Line 344: why not 3-6 weeks which has been previously described as the time frame for band castration to take place?
Line 345-364: the authors should redirect their focus in this paragraph from criticising the existing measures (which were assessed by the group) to proposing which are the appropriate pain assessment methodology for this type of studies. Showing the results from the other variables could help the authors to describe where are the limitations of the mentioned parameters and which variables could be used instead.
Line 371: there was only 1 behavioral and 1 physiological parameter assessed in this study.
Line 373: please describe the ‘backbone of fundamental pain observation tools’ as it is not clear as to what the authors are referring to.
Line 374: reword, as electrostimulation does not evaluate delivery and function of lidocaine it only assess sensitivity.
Line 375: what time period is considered acute and chronic?
Line 375: what parameter was used to test these regions? In materials and methods electrostimulation is described to be used only above the band.
Line 392: cortisol was never mentioned.
The authors should discuss how these results compare to other studies assessing anesthetics at the time of castration and how the results compare to other in vitro studies?